# Epigenetic Signatures of Social Defeat Stress Varying Duration

**DOI:** 10.3390/ijms27010018

**Published:** 2025-12-19

**Authors:** Natalya Bondar, Vasiliy Reshetnikov, Polina Ritter, Nikita Ershov, Natalia Zhukova, Semyon Kolmykov, Tatyana Merkulova

**Affiliations:** 1Institute of Cytology and Genetics, Lavrentiev Ave., 10, Novosibirsk 630090, Russia; 2Department of Natural Sciences, Novosibirsk State University, Pirogova St., 2, Novosibirsk 630090, Russia; 3Translational Medicine Research Center, Sirius University of Science and Technology, 1 Olympic Avenue, Sochi 354340, Russia

**Keywords:** social defeat stress, depression, prefrontal cortex, H3K4me3, ChIP-seq, epigenetics

## Abstract

Stress-induced mental disorders, including depression and anxiety disorders, constitute a global issue in contemporary society due to treatment complexity and the diversity of manifestations. Understanding the molecular mechanisms of these disorders presents a significant challenge for neurobiology. We investigated the effects of social defeat stress (SDS) of varying durations (10 and 30 days) on behavioral patterns and the H3K4me3 (trimethylation at the 4th lysine residue of histone H3) landscape in the prefrontal cortex of C57BL/6 mice. Furthermore, we compared these data with previously published H3K4me3 landscape data obtained after 15 days of SDS and transcriptomic data collected after 10, 15, and 30 days. We discovered that a 30-day period of stress results in more pronounced depressive-like behavior. SDS induces slight alterations in the H3K4me3 density across numerous nucleosomal peaks. The analysis of differential enrichment peaks of H3K4me3 in promoter regions following varying durations of SDS revealed that the aggregation of multiple H3K4me3 nucleosome peaks in the promoter region functions as a QR code, likely affecting the promoter’s state regarding the accessibility of transcription factors. Furthermore, we identified a cluster of genes in the promoter regions exhibiting differential enrichment peaks of H3K4me3 following SDS of any duration. This cluster includes genes encoding transcription factors such as *Mef2c* and *Nr4a3*, as well as postsynaptic density proteins (*Shank2*, *Shank1*, and *Gria2*), which are associated with stress sensitivity and the onset of depression; their protein products are involved in synaptic transmission and signal transduction mechanisms. The comparison of ChIP-seq and RNA-seq data following varying durations of SDS enabled a deeper insight in to the dynamics of SDS-induced changes. Together, these findings provide a better understanding of the molecular mechanisms of SDS in the prefrontal cortex.

## 1. Introduction

Stress is a primary risk factor for prevalent psychopathologies, including major depression and generalised anxiety disorder [1,2]. It also increases the risk of many metabolic [3,4], cardiovascular [5,6], and neurodegenerative [7] diseases. The extensive negative consequences of chronic stress are associated with alterations in the brain’s molecular profile and neurochemistry, which regulate behavioral and physiological responses, directly affecting the metabolic, cardiovascular, and immune systems [8]. Chronic stress is considered “toxic stress” and can lead to a decrease in brain plasticity and the inability of the neural circuitry to adapt to a new situation. Such adaptations at the molecular level are driven by changes in gene expression and epigenetic modifications [9].

Despite considerable advancements in the investigation of stress-induced psychopathologies over the last twenty years, largely driven by the emergence of high-throughput next-generation sequencing techniques and precise optogenetic methodologies, the molecular mechanisms underlying their development remain poorly understood. Various rodent stress models, including social isolation, social instability, chronic variable stress, learned helplessness, and chronic social defeat stress (chronic SDS), are employed to investigate the molecular mechanisms of depression [10].

Among these models, chronic social defeat stress (SDS) is the most ethologically valid animal model of depression, inducing a depression-like state in mice [11]. Numerous studies have been undertaken to evaluate transcriptomic alterations following chronic SDS in adult male mice across different brain structures [12,13,14]. However, because transcriptomic alterations are highly sensitive to various methodological factors, only a restricted number of genes demonstrated consistent unidirectional changes across multiple studies [15]. The majority of studies on brain epigenetic modifications caused by SDS have focused on either examining the regulatory regions of specific genes or evaluating the overall levels of certain histone marks through semiquantitative techniques [16,17,18]. To comprehend the molecular mechanisms of chronic SDS, a genome-wide evaluation of epigenetic alterations in gene regulatory regions is essential. The number of comparable studies is quite restricted; only recently has genome-wide differential methylation of DNA been evaluated after 10 days of SDS in the hippocampus, nucleus accumbens and the whole brain [19,20,21]. Our group previously performed the study assessing the effects of a 15-day chronic SDS on the genome-wide H3K4me3 landscape in the prefrontal cortex of C57BL/6 mice subjected to maternal separation early in life [15]. H3K4me3 is generally located in the promoter region of genes and is a marker indicative of an open chromatin configuration and active gene transcription [22]. Changes in the H3K4me3 pattern may signify the molecular adaptation of prefrontal cortex cells to chronic stress [15].

This study examined the impact of social defeat stress of different durations (10 and 30 days) on the genome-wide H3K4me3 landscape in the prefrontal cortex of C57BL/6 mice. These durations were selected for comparison with a prior study on transcriptomic alterations in the prefrontal cortex [23]. This region is essential in modulating responses to stress, as well as anxiety and depression-like behaviors induced by chronic stress [24,25,26,27]. The duration of SDS affects both transcriptomic changes in the prefrontal cortex and depressive-like behavior in C57BL/6 mice [15,23]. Although we have previously demonstrated that the density of H3K4me3 in gene promoters does not correlate with gene expression [28], we hypothesise that the H3K4me3 landscape may be selectively modified in genes involved in stress response and depression. Furthermore, to investigate the relationship between these alterations in gene expression dynamics, we compared the genome-wide H3K4me3 landscape changes in mice following varying durations of SDS with transcriptomic data obtained in a similarly designed experiment.

## 2. Results and Discussion

### 2.1. 30 Days of SDS Causes More Pronounced Behavioral Disturbances Compared to 10 Days of SDS

We evaluated the effects of chronic social defeat stress of varying durations on sociability/social avoidance and depression-like behavior using the partition test and the forced swim test (Figure 1a). Furthermore, we also assessed the adrenal index as a measure of alterations in the hypothalamic–pituitary–adrenal axis induced by SDS. SDS influenced the duration spent near the partition, approach latency, immobility duration and latency in forced swim test (Kruskal–Wallis test: H (2, 41) = 18.8, *p* < 0.001; H (2, 39) = 12.56, *p* < 0.01; H (2, 41) = 8.38, *p* < 0.05; H (2, 41) = 12.55, *p* < 0.01). Mice in the S30 group showed reduced latency to the initial immobility phase and increased immobility duration in the Forced Swim Test (FST) compared to the control group (*p* < 0.01), indicating the emergence of depressive-like behavior (Figure 1b).

Mice in the S10 group did not differ from controls on these measures. Animals from both stressed groups (S30 and S10) showed a diminished response to the partner in the partition test relative to the control group (*p* < 0.05). Together, these results demonstrate more pronounced behavioral disturbances in the S30 group relative to S10.

Each mouse was assigned a behavioral score based on the time spent near the partition and the duration of immobility. Values between 2 and 3 signify “susceptible” animals whereas values above 4 indicate “resilient” animals (Figure 1c). Our findings indicate that the majority of mice in the S30 group exhibited susceptibility, whereas approximately half of the S10 group were classified as “susceptible” with the remainder categorized as “resilient”. The data align with other studies [23,29] indicating that 21 or 30 days of social defeat stress lead to distinct depression-like characteristics, such as heightened social avoidance, increased immobility in the forced swim test, and anhedonic behavior, in contrast to animals subjected to 10 days of social defeat stress. After 10 days of social defeat stress (SDS), some animals (“susceptible animals”) show behavioral alterations characterised by social withdrawal and heightened anxiety, while others display only minimal changes (“resilient animals”) [30,31]. Nonetheless, the segregation of these groups, typically conducted according to the social interaction index results, may be unsuitable [32,33]. The heterogeneity observed after 10 days of social defeat stress may reflect different adaptive strategies employed by animals, influenced by their individual characteristics prior to the stressor [32]. However, prolonged stress results in more pronounced behavioral disturbances in the S30 group relative to S10, potentially linked to both the duration of the stress exposure and a reduction in heterogeneity among the animals due to the adoption of a singular adaptation strategy under extended stress conditions.

### 2.2. SDS Alters Density of Nucleosome H3K4me3 Peaks Depending on the Stress Duration

We identified 153,203 nucleosome-sized (147 bp) H3K4me3 peaks (Figure 2A). Most (71.5%) of the identified peaks were located within ±2000 bp from the transcription start site (TSS) of genes, corroborating previous research on the genomic distribution of H3K4me3 peaks.

To biologically interpret the ChIP-seq results, we focused on peaks located within ENCODE candidate cis-regulatory elements (cCREs). cCREs are a subset of representative DNase hypersensitive sites from ENCODE samples, validated by either histone modifications (H3K4me3 and H3K27ac) or CTCF-binding data.

cCREs encompass promoter-like signatures (PLSs, ±200 bp around the TSS), proximal enhancer-like signatures (pELS, ±2000 around the TSS), distal enhancer-like signatures (dELSs, >2000 around the TSS), DNase-H3K4me3 cCREs, and CTCF-only cCREs. We assigned each cCRE to the nearest TSS of known transcripts and retained only those associated with genes expressed in the mouse prefrontal cortex (based on our previously acquired data [23], threshold FPKM > 0.1). The resultant set of peaks, designated cCRE, was employed to evaluate disparities between experimental groups using DESeq2.

Analysis of the H3K4me3 landscape showed no significant differences between groups after correction for multiple comparisons (Benjamini–Hochberg correction, adjusted *p* < 0.1), with the exception of 8 peaks that were downregulated in the S30 group relative to the S10 group (Table 1). These peaks are located in the regulatory regions of genes *Reck*, *Mmp17*, *Fbxo4*, *Pan3*, *Prickle1*, *Tmod3*, *Arid1b*, *Tmem165*. The gene *Arid1b*, which encodes a protein that is part of the SWI/SNF chromatin remodeling complex, is particularly noteworthy in relation to the mechanisms of stress and depression, as it may play a role in cell-cycle activation. The downregulation of *Arid1b* expression disrupts cortical interneuron development and mouse behavior, and correlates with neurodevelopmental and psychiatric disorders in humans [34,35].

Consequently, we examined data employing relaxed thresholds for a nominal *p*-value of less than 0.05 (hereinafter referred to as DE peaks). We identified variations in H3K4 methylation across 2223 peaks between the S10 group and the control, situated within 1866 genes (Figure 2B, Appendix A). A 30-day stress period led to alterations in 2556 peaks relative to the control group, encompassing a total of 2227 genes. In the comparison of groups experiencing varying durations of stress (S30 versus S10), 3657 differentially expressed peaks were identified in proximity to 2888 genes. The majority of DE peaks in the S30 group exhibited a lower density relative to the control (63% of peaks) and the S10 group (approximately 66%). In the S10 group, most peaks (60%) showed higher density than in controls (Figure 2C). Only a small subset of peaks was altered in both groups (15.7% increased and 7.3% decreased), and all shared peaks showed changes in the same direction. In the peaks that are not classified as cCREs, variations were observed based on stress duration: 2421 DE peaks in the S10 group, 2578 DE peaks in the S30 group, and 3239 DE peaks when comparing the S30 and S10 groups (Appendix A). This suggests potential functional relevance of these regions. However, due to the intricacy of associating these peaks with specific functional elements of the genome, we will focus exclusively on peaks situated within cCRE.

Alterations in gene transcription within the prefrontal cortex mirror the direction of peak changes: after 10 days most genes were upregulated, whereas after 30 days they were predominantly downregulated [23]. Nonetheless, we found no correlation between the expression levels of the genes associated with these peaks and the variation in the density of individual peaks. Our analysis focuses on individual nucleosomal peaks, revealing that expression levels are contingent upon the interplay of upregulated and downregulated peaks across various regulatory elements of the gene. The heightened density of peaks in the promoter may indicate increased promoter activation potential.

Although many peaks were altered, the magnitude of change was generally modest. The mean ‖log2FoldChange‖ for DE peaks between groups was 0.52, and only 1.4% of all DE peaks showed ≥2-fold changes. A similar lack of pronounced changes in epigenetic landscapes was also shown in other genome-wide epigenetic studies after 10–14 days of SDS, in which the level of DNA methylation in the dentate gyrus of hippocampus (Methyl-Seq) and nucleus accumbens (MethylC-Seq) was assessed [20]. One of the reasons for the absence of strong epigenetic changes may be that bulk prefrontal cortex tissue was used in our and other studies. Bulk H3K4me3 signals may obscure cell-type-specific changes and mask significant alterations that are present in individual cell types [36]. The problem of mixing signals from different types of cells can be solved by preliminary cell sorting; however, a long process of sample preparation and additional incubation with antibodies can alter epigenetic marks, especially on unfixed tissues [37]. The problem of isolating cell type specific signals from bulk data is also solved using bioinformatics approaches. Thus, several analysis methods using single-cell sequencing data have been developed to analyze RNA-seq data [38,39]. Similar approaches have been developed for DNA methylation, accounting for the contributions of different cell populations and their methylation levels, thereby enabling bulk data to be deconvolved to single-cell resolution [40,41]. The emergence of similar algorithms for analyzing the distribution of chromatin modifications would make it possible to detect significant changes with greater accuracy.

### 2.3. Distribution of Peaks Among cCRE Regulatory Elements

We examined the regulatory elements affected by peak-density changes linked to different stress durations (Figure 2D). The distribution of DE peaks among cCRE elements indicates that the proportions of up- and down-regulated peaks in proximal pELS enhancers were similar across groups. Nonetheless, up-regulated peaks are predominant in the PLS promoter elements of the S10 group, whereas down-regulated peaks are predominant in the promoters of the S30 group. The opposite pattern was observed in distal dELS enhancers. The lengths of ENCODE cCREs vary from 150 to 350 bp, comparable to the length of a single nucleosome peak (147 bp); thus, 94% of cCREs contain a single DE peak, 5.6% (362 cCRE) exhibit two DE peaks, and merely 16 cCREs (0.2%) contain three DE peaks, most of which showed changes in the same direction. Only 7 cCREs (1.9% of all cCREs with multiple DE peaks in an element) exhibit peaks where the H3K4me3 levels fluctuate in opposing directions. Thus, peak-density variation generally reflects the overall change in the corresponding cCRE.

We examined the patterns of cCRE alterations across stress durations (Figure 3). In the S10 group, 73% (264 peaks) of PLSs were up-regulated and 70% (128 peaks) were down-regulated, while in the S30 group, the activity level returned to that of the control group with prolonged stress duration. Following prolonged stress, new down-regulated PLSs appeared, while only 108 new up-regulated PLSs were observed. Only 108 PLSs showed H3K4me3 changes that were independent of stress duration.

Consequently, as stress duration increases, promoter activity shifts, with a higher proportion of elements showing reduced activity (from 36.6% to 78.9%, approximately 90% of which are newly identified downregulated PLSs).

A similar pattern was observed for proximal enhancers: In the S10 group, 75.5% (584) of pELS were up-regulated and 70% (388) were down-regulated, returning their activity to control levels with prolonged stress duration in the S30 group. In the S30 group, other pELS were altered: 459 were down-regulated and 319 up-regulated. Only a small subset of enhancers (190 up- and 103 down-regulated pELS) maintained their activity regardless of stress duration. Consequently, the proximal enhancers showed change patterns similar to promoters, although in a more gradual manner. The proportion of down-regulated pELS increased from 38.8% to 73.3%, ~88% of which were newly down-regulated.

A markedly different pattern was observed for distal enhancers (dELSs). Social defeat stress over a 10-day period resulted in equivalent numbers of up-regulated and down-regulated dELSs, with 162 dELSs upregulated and 168 dELSs downregulated, respectively. Of these, 68% of up-regulated and 90% of down-regulated dELSs are returned to baseline levels. As the duration of stress increases, 121 novel up-regulated dELSs and only 50 novel down-regulated dELSs emerge. Consequently, prolonged stress increased the proportion of dELSs with elevated activity (from 50% to 72%), 73% of which were newly up-regulated).

Notably, DNase-H3K4me3 cCREs, along with PLSs and pELSs, showed a reduction in the percentage of down-regulated cCREs in the S30 group relative to the S10 group (from 60% to 8%). Identifying patterns for CTCF-only is unfeasible due to the limited number of cCREs exhibiting DE peaks, totalling 13 cCREs.

Thus, promoters showed more extensive alterations, predominantly decreases in activity. The number of proximal enhancers is comparable, yet inhibitory alterations also prevail. Fewer distal enhancers were altered, but their activity, in contrast, tended to increase.

### 2.4. Gene Ontology Analysis Links Chromatin Dynamics to Specific Biological Processes Across Stress Durations

Changes in H3K4me3 levels can differentially affect gene activity through their impact on chromatin accessibility. Consequently, we classified all DE peaks as up- or down-regulated (Figure 2C) and analysed the genes associated with peaks showing consistent alterations across groups (Figure 4).

Axon guidance and morphogenesis are a shared GO-enriched category for genes with up-regulated peaks in both the S10 and S30 groups. However, up-regulated peaks specific to the S10 group were identified in regulatory elements of genes primarily linked to transcription regulation, including transcription factors (*Nfib*, *Etv1*, *Pax6*, *Mef2c*, *Tbr1*, and *Nr4a3*) and adhesion molecules (*Robo3*, *Trio*, *Alcam*). Conversely, up-regulated peaks specific to the S30 group were predominantly associated with genes involved in semaphorin receptor activity and signaling, including *Sema6c*, *Sema6b*, *Sema7a*, *Plxd1*, *Plxna2*, and *Nrp1*, as well as cell adhesion molecules such as *Ank3*, *Nfasc*, *Gap43*, and *Nrcam*. Semaphorin signaling pathway may contribute to pathological changes in neuronal connectivity characteristic of chronic stress, including synaptic reorganization, synapse loss, and altered synaptic function [42,43]. Thus, prolonged stress engages genes that regulate the refinement of synaptic connections.

The up-regulated peaks specific to the S30 group are mapped to the regulatory elements of 596 genes and, unlike the S10 group, form tight functional clusters. The enriched GO terms were primarily related to synaptic organisation. These epigenetic changes, which manifest only after prolonged stress, appear to amplify the disruptions from earlier stages. A key finding is the increased chromatin accessibility within regulatory elements of genes essential for synaptic integrity, including those encoding postsynaptic density proteins (*Shank2*, *Dlgap3*, *Syngr1*, *Lrrc7*) and cell adhesion molecules (*Nrcam*, *Nfasc*, *Mdga1*, *Nptn*, *Cadm2*, *Opcml*, *Ptprs*, *Ptprz1*, etc.). This coordinated chromatin opening does not necessarily lead to increased gene expression but rather suggests that the genome is being primed for a state of hyper-plasticity, creating a permissive environment for synaptic remodeling [44].

Decreased chromatin accessibility at down-regulated peaks specific to the S10 group was mapped to regulatory elements of 663 genes, which were enriched for growth factor response, epithelial cell differentiation, Wnt signalling, and gliogenesis. This repressive pattern was markedly amplified after prolonged stress. In the S30 group, the number of affected genes nearly doubled to 1259, with a dramatic shift in the localization of these peaks towards promoter elements. These genes were clustered across 49 uniquely enriched categories—distinct from all other groups—including protein ubiquitination, RNA stability, protein kinase activity, and chromatin organisation. Together, these findings point to a pronounced and escalating imbalance in the chromatin landscape, characterized by a widespread suppression of regulatory potential, particularly at gene promoters, under conditions of chronic stress.

This dysregulation was most evident in the marked down-regulation of peaks associated with a set of 127 genes involved in chromatin organisation and histone binding, primarily affecting their promoters and proximal enhancers. In stark contrast, up-regulated peaks were observed for only 39 genes and were predominantly localized to enhancers. Importantly, the reduction in peak density affected the regulatory elements of key transcriptional activators, including histone acetyltransferases (Crebbp, Kat2b), histone methyltransferases (Nsd1, Setd2), the lysine-specific demethylase Kdm6b (which removes the repressive H3K27me3 mark), and the methylcytosine dioxygenase Tet2, involved in DNA demethylation. Furthermore, reduced peak density was also observed for genes encoding the H3K4 trimethylation enzymes Setd1a and Kmt2c, a finding accompanied by decreased expression of these genes [23]. Since these enzymes deposit the H3K4me3 activation mark, their diminished expression provides a plausible mechanism for the accumulation of down-regulated peaks over time. However, this pattern does not reflect a simple global repression of transcriptional activity, as down-regulated peaks were also detected in the promoters of genes involved in transcriptional repression (e.g., Kdm2a, Kdm3b, Sirt1, Suv39h1). Together, these findings reveal a complex, bidirectional dysregulation of the transcriptional machinery under prolonged stress, rather than a unidirectional silencing.

### 2.5. Motif Enrichment Analysis

Motif enrichment analysis at nucleosome peak summits revealed a significant enrichment of transcription factor binding sites in up-regulated peaks across both experimental groups (S10–41 TF, S30–34 TF, 24 TF common to both groups), suggesting potential activation of the transcriptional programme and epigenetic reprogramming in response to stress. Key factors related to neuronal plasticity, stress adaptation, and regulation of neuroinflammation include Nr4a1, Mef2a, Stat1, and Irf8, whose binding sites were enriched in up-regulated peaks in both experimental groups (Figure 5A). Mef2a is integral to neuronal plasticity, synaptogenesis, and cellular survival in the brain [45] and facilitates axonal regeneration in traumatic brain injury [46]. Reduced Mef2a function results in a decrease in dendritic spines and a deterioration of synaptic connections, contributing to stress-related depression. Two additional factors, Stat1 and Irf8, are associated with neuroinflammation [47]. In particular, Stat1 regulates intrinsic programs in neural stem cells (NSCs) during neuroinflammation [48], participates in myelination [49], suppresses axon degeneration and neuronal cell protection [50]. Notably, the results of human and animal studies highlight the critical role of oligodendroglial and myelin-related aberrations in stress-related psychiatric disorders [51]. The key regulator identified was Nr4a1 (Figure 5B), an early response gene [52], whose expression is rapidly and transiently induced in the central nervous system by various stimuli. Nr4a1 contributes to learning and memory functions, as well as neuroprotection by inhibiting apoptosis and oxidative stress [53]; dopaminergic signalling [54]; memory formation and inhibiting neuroinflammation [55]. Additionally, Nr4a1 plays a role in the differentiation of glucocorticoid-resistant, pathogenic Th17 lymphocytes, which are implicated in the pathogenesis of depression [56].

Among the transcription factors with enriched binding sites exclusively in the up-regulated peaks of the S10 group, Foxo3, a member of the Foxo transcription factor family, warrants emphasis. Foxo3 is expressed in critical brain regions and plays a significant role in neuronal responses to external stimuli, particularly in regulating individual behavior, oxidative stress responses, and neuron-mediated cognitive impairment [57,58,59]. Among the transcription factors with binding sites exclusively enriched in up-regulated peaks within the S30 group, it is important to emphasise Heat shock factor 1 (HSF1) is a principal transcriptional factor responsive to stress, safeguarding cells from apoptosis through both coding and non-coding responses (lncRNAs) [60]. Specifically, HSF1 promoter-specific control of Bdnf gene regulation plays an important role in neuronal protection and plasticity in the hippocampus in response to acute stress [61]. Down-regulated peaks showed far fewer enriched transcription factor motifs. Enrichment of Egr4, Mecp2, and Mbd2 sites was observed solely among the down-regulated peaks in the S30 group (Figure 5C,D). Methyl-CpG binding domain 2 (Mbd2) and methyl-CpG binding protein 2 (Mecp2) are chromosomal proteins, that bind to methylated CpG sites on DNA and play key role in epigenetic regulation by repressing gene transcription [62,63]. These proteins participate in synaptic maturation, plasticity, cognition, and experience-dependent epigenetic programming [62,64,65]. Previous studies report excessive activation of MeCP2 is observed due to chronic stress and is linked to the onset of depression [65]. A proposed molecular mechanism involves the overexpression of MeCP2, which can suppress BDNF transcription, leading to the downregulation of BDNF mRNA and protein levels [66]. In our study, MeCP2 motifs were enriched in the down-regulated peaks exclusively in the S30 group, but not in the S10 group. This finding is consistent with the increased number of down-regulated peaks in this group and the identified reorganization of genes involved in chromatin organization. Regarding the transcription factor Egr4, which exhibits a neural-specific expression pattern [67,68] and is primarily associated with gene activation, the detection of its binding sites within down-regulated peaks indicates a diminished response of its target genes to activating stimuli. We propose that Nr4a1 can influence its targets irrespective of stress duration, while the epigenetic reprogramming of MeCP2, Mbd2 and Egr4 are initiated solely by extended social defeat stress (in the S30 group).

### 2.6. The Aggregation of Numerous H3K4me3 Nucleosome Peaks in the Promoter Region Likely Delineates Distinct Signatures for Transcription Factor Accessibility and May Affect Expression Levels

Of the 671 genes with altered peak density across both experimental groups, the peaks were consistent in the majority of genes, exhibiting uniform directional changes compared to the control: 410 genes shared one DE peak, 27 genes shared two, and 5 genes shared three identical DE peaks in S10 and S30 groups.

Only 229 genes (34.1%) in the S30 and S10 groups exhibited distinct alterations in DE peaks. Our findings reveal that a substantial proportion of genes showed DE peaks demonstrated variability in the localisation and quantity of DE peaks within the promoter region of individual genes, contingent upon the duration of stress exposure. Dynamic changes in the density of nucleosome-size peaks at each promoter facilitate precise regulation of gene expression, as recent studies suggest that H3K4me3 may contribute to both the activation and repression of specific genes. At the molecular level, the distinctive arrangement of H3K4me3 peaks may influence promoter accessibility to transcription factors and the pattern of gene expression.

Subsequently, among genes with shared DE peaks in S10 and S30, only one peak (chr11: 96,064,898–96,065,045) differed between groups, located in the promoter of the ubiquitin-conjugating enzyme *Ube2z*. The density of H3K4me3 in this peak diminished with prolonged stress duration. Nonetheless, as these alterations were noted in a single gene, it suggests that the impact of stress of differing durations is predominantly independent of the density of individual peaks, but rather contingent upon the distinct profile of multiple H3K4me3 peaks within the promoter region.

To identify the most consistent SDS-related changes, we compared ChIP-seq results from S10 and S30 with those acquired after 15 days of SDS (S15), as reported in previous studies [15]. Only six peaks located in *Numbl*, *Gal3st3*, *Cntn2*, *Shank1*, *Pcdh10*, and *Gria2* showed the same direction of change across all three groups, and the extent of these alterations was independent of the stress duration (Appendix A). In the 73 genes exhibiting at least one differentially expressed peak, the H3K4me3 landscape varied among groups depending on SDS duration, resulting in a distinct peak-distribution “QR code” (Figure 6) that likely shapes promoter accessibility to transcription factors.

Ultimately, we correlated alterations in H3K4me3 levels in the prefrontal cortex with gene expression levels derived from RNA-seq data [15,23] (Appendix A). Only the *Pde10a* gene, which encodes the phosphodiesterase-10A enzyme, exhibited differential expression peaks in its promoter and altered expression across all groups (Figure 6). *Pde10a* governs the accumulation of cAMP and cGMP and modulates intracellular signalling pathways [69]. PDE10 is linked to the onset of neurodegenerative and psychiatric disorders, such as major depressive disorders [70,71,72], rendering it a compelling therapeutic target.

A small proportion of DE genes (9–16%) had DE peaks in their promoter regions. These data are consistent with our previous findings from RNA-seq and ChIP-seq analyses performed on the prefrontal cortex, suggesting that modifications in the epigenetic landscape after SDS did not correlate with gene expression levels [28]. H3K4me3 interacts with the transcriptional machinery through chromatin modification/remodelling and by directly influencing components of the transcriptional apparatus [73]. The impact of perturbing H3K4me3 on gene expression is complex, and its regulatory function in transcription remains debated [22,73]. Elevated H3K4me3 density may result from transcription, affecting processes like splicing and transcription termination [22]. Recent findings show that the progression from H3K4me1 to H3K4me2 and subsequently to H3K4me3 transpires incrementally across several transcription cycles, with nucleosome retention; heightened transcription frequency and consistency would facilitate the dissemination of H3K4me3 [74]. Thus, elevated H3K4me3 density is a result, rather than a precursor, of heightened expression.

To assess whether alterations in H3K4me3 density are a cause or effect of changes in gene expression, we examined the proportion of differentially expressed (DE) genes with DE peaks in their promoters to the total number of DE genes across stress durations [15]. We observed that alterations in peak density after 10 days of stress exhibited a significantly stronger correlation with changes in gene expression after 15 and 30 days of stress than with changes in expression after 10 days (Table 2, 15.74% in S15 and 12.01% in S30 vs. 9.08% in S10, *p*(χ^2^) < 0.05). This supports the hypothesis that alterations in H3K4me3 density may precede subsequent changes in gene expression. These conclusions are primarily speculative and do not account for several factors, including the direction of change, the impact of alterations in other peaks, and experimental bias.

The study’s limitations also encompass the restricted number of animals utilised for ChIP-seq analysis. Additionally, there is an absence of orthogonal methods to validate the identified molecular alterations. Furthermore, the epigenetic landscape extends beyond the H3K4me3 modification. Consequently, to comprehend the intricate molecular mechanisms underlying stress-induced behavioral disturbances, future targeted studies examining specific gene networks are necessary.

## 3. Materials and Methods

### 3.1. Animals

C57BL/6J and CD1 mice, aged 10–12 weeks, were maintained at the Centre for Genetic Resources of Laboratory Animals, Institute of Cytology and Genetics (SB RAS, Novosibirsk, Russia, RFMEFI62119X0023). The animals were maintained under standard conditions (12:12 h light/dark cycle, with illumination commencing at 8:00 a.m.); feed pellets and water were provided ad libitum.

### 3.2. Experimental Design

Chronic social defeat stress was induced following the modified protocol of the sensory contact model [75], as previously described [15,23]. A C57BL/6 mouse was positioned in an empty section of a steel cage (14 × 28 × 10 cm) next to an aggressive CD1 mouse, separated by a perforated transparent barrier. The animals acclimated to the new housing conditions for 2 days before undergoing chronic SDS for either 10 consecutive days (S10 group) or 30 consecutive days (S30 group). The C57BL/6 mouse under investigation was subjected to an aggressive assault by a CD1 mouse for 10 min daily, while remaining separated from the CD1 mouse by a partition during the remainder of the time. To mitigate individual-specific effects, each C57BL/6 mouse was placed in an unfamiliar cage with a new aggressive CD1 mouse behind a partition once daily, following the defeat session. Control animals were also housed in a steel cage with a partition three days prior to the initial behavioral test, but without an aggressive CD1 mouse positioned behind the partition. During days 8–9 of social defeat for group S10 and days 28–29 for group S30, behavior was assessed through the “partition” and forced swim tests. The animals were put to death by decapitation twenty-four hours after the last social defeat session, and the prefrontal cortex was carefully dissected as previously reported [15]. The adrenal glands were weighted and adrenal indices were calculated (adrenal mass (ng)/body mass (g)). The prefrontal cortex were promptly frozen in liquid nitrogen and preserved in 1.5 mL plastic tubes at −80 °C for subsequent analysis.

### 3.3. Partition Test

The partition test was used to evaluate sociability and social avoidance [76]. The test was conducted in an experimental cage divided by a permeable, perforated partition. There was a CD1 aggressor mouse on one side of the partition and a study mouse on the other. As markers of responding to the partner, the amount of time spent close to the partition and the latency of the initial approach were scored over a 5-min period.

### 3.4. Forced Swim Test

The forced swim test was employed to evaluate depression-like behavior [77]. Mice were placed individually into a glass cylinder measuring 30 cm in height and 10 cm in diameter. cm) containing water (19 cm deep, 23–24 °C). Immobility latency and total time of immobility (floating) were evaluated during a 5 min test period.

### 3.5. Evaluation of Behavioral Score

The values for duration spent near the partition and duration of immobility were divided into 4 quartiles, each assigned a score from 1 to 4 (1—the shortest time approaching the partition and the longest immobility time; 4—the longest time approaching the partition and the shortest immobility time). The sum of scores from the two behavioral tests reflected the depressive score of the mice, based on which we selected animals for sequencing.

### 3.6. Chromatin Immunoprecipitation and Library Construction

Mice from each cohort (control—3 samples; S10 group—5 samples; and S30 group—5 samples) for chromatin immunoprecipitation. Animals exhibiting the most characteristic behavior for each group were chosen for immunoprecipitation. In each test, the animal was assigned a rank based on its behavior. The ranks were aggregated for each animal. Animals exhibiting the most disparate group ranks were chosen for sequencing. Immunoprecipitation utilising an antibody against H3K4me3 (ab8580, Abcam, Cambridge, UK) was conducted employing the native immunoprecipitation technique [78] with certain modifications as previously outlined [15,79]. Employing micrococcal nuclease (MNase) (NEB, Ipswich, MA, USA, #M0247S), which preferentially preserves nucleosome-bound genomic DNA, enables the acquisition of peaks with nucleosome-size resolution [80]. Briefly, samples of the prefrontal cortex were homogenized in douncing buffer (10 mM Tris-HCl pH 7.5, 4 mM MgCl_2_, 1 mM CaCl_2_) containing Halt Protease Inhibitor Cocktail (Thermo Scientific, Waltham, MA, USA) and homogenate was incubated with 2000 U of MNase for 6 min on 37 °C, 700 rpm. Following the cessation of the reaction through the addition of 10 mM EDTA, each sample was incubated in a fourfold excess of hypotonic lysis buffer (0.2 mM EDTA, 1.5 mM DTT, Halt Protease Inhibitor Cocktail) at 4 °C for 60 min with rotation, after which debris was eliminated via centrifugation at 3000× *g*. To diminish background contamination, chromatin DNA samples were precleared by incubating with G protein-conjugated magnetic beads (NEB, #S1430S) for 2 h, followed by overnight incubation of the purified samples with H3K4me3 antibody pre-immobilized on G protein magnetic beads. Ten percent of each precleared sample was retained as input DNA for subsequent ChIP enrichment analysis.

Following incubation, beads containing the captured H3K4me3-associated chromatin complexes were subjected to washes with Low Salt, High Salt, LiCl, and TE buffers (details provided in [79]) before eluting chromatin DNA samples from the beads. Samples and input DNA underwent treatment with proteinase K and RNase A, followed by purification via phenol-chloroform extraction. Quantitative PCR analysis with specific primers was conducted as previously described to analyse ChIP enrichment [15,79]. Only samples exhibiting fold enrichment greater than 25 were utilised for library preparation.

ChIP-seq libraries were constructed using the NEBNext Ultra II DNA Library Prep Kit for Illumina. The selection of DNA fragment sizes was conducted using Agencourt AMPure XP beads (Beckman Coulter, Brea, CA, USA). Subsequently, PCR enrichment of the adapter-ligated library was performed, comprising six to eight cycles of PCR. The dimensions and amounts of each library were assessed using an Agilent Bioanalyzer 2100. Single-end sequencing (1 × 100 bp) for ChIP-seq was conducted on the Illumina HiSeq 4000 platform (Evrogen Joint Stock Company, Moscow, Russia).

### 3.7. Analysis of ChIP-seq Data

Each sample yielded an average of approximately 38.2 million unpaired reads, ranging from 22.5 to 47.9 million. The sequencing data were preprocessed using the Trimmomatic tool [81] for adapter removal and subsequently aligned to the GRCm38/mm10 mouse reference genome with the bowtie2 aligner [82]. The quality metrics of ChIP-seq libraries (Appendix A) were evaluated using the Phantompeakqualtools software (version 2.0) [83]. The MACS2 algorithm, utilising nucleosome-optimized parameters (--shift 37 --extsize 73), was employed to identify single-nucleosome peaks in the pooled data compared to a micrococcal-nuclease-treated input library. The nucleosome-size peak is defined as a 147 bp sequence, encompassing the coordinates of single-nucleosome peak summits (1 bp) along with 73 bp sequences situated both upstream and downstream of it. A sample from one control mouse was utilised as the background in the peak calling procedure.

The DESeq2 R package (version 1.44.0) was employed for differential enrichment (DE) analysis. In the DESeq2 workflow, reads with a mapping quality of no less than 10 were quantified for each peak region. Peaks that overlapped with the DAC blacklisted regions (ENCODE accession No. ENCFF547MET) were omitted from the subsequent analysis. Normalisation, modelling, and statistical testing were conducted using default parameters. The peaks exceeding the *p*-value of <0.05 were classified as significantly differentially enriched peaks (DE peaks) for subsequent analysis. The identified H3K4me3 peaks were allocated to genes according to the nearest annotated transcription start site. The GENCODE comprehensive gene annotation release M13 (GRCm38.p5) was utilised to associate the predicted peaks with the nearest transcription start sites of genes.

All steps of the described data-processing pipeline were implemented as an automated workflow for a new bioinformatics platform being developed by Sirius University (https://github.com/genespace-ru/workflow-engine, accessed on 14 December 2025). Developed workflows are available at repository https://repository.genespace.ru.

### 3.8. Motif Enrichment Analysis

Motif analysis was performed using the AME tool of the MEME Suit implemented in R library memes (v1.0.4). For the analysis we used the HOCOMOCOv11 mouse TF database. Using our previously obtained PFC RNA-seq data we subsetted the database for TFs expressed in mice PFC resulting in 278 PWMs used as AME database input. For the analysis we used DNA sequences from DE single-nucleosome peak summits ± 73 bp (Appendix A). As background sequences of all single-nucleosome peaks in cCRE were used (summits ± 73 bp).

### 3.9. GO Enrichment Analysis

We performed Gene Ontology (GO) enrichment analysis on a target set of genes defined by two criteria: (1) association with a cCRE harboring differentially enriched (DE) peaks, and (2) expression in the prefrontal cortex (PFC). The analysis used as a background set all genes expressed in the PFC that were associated with any cCRE. We defined PFC-expressed genes using our previously published dataset ([23]; NCBI BioProject PRJNA323485), applying an expression threshold of FPKM > 0.1 (17,680 genes).

GO enrichment analysis was performed using the clusterProfiler package. Significantly enriched terms were defined as those with an FDR < 0.05. To focus on more interpretable categories, we excluded GO terms containing more than 500 genes from the reported results, as these represent overly broad biological processes.

### 3.10. Using Public Available RNA-seq and ChIP-seq Data

To extend our analysis of H3K4me3 dynamics, we integrated our previously published datasets. This included H3K4me3 ChIP-seq data from a 15-day SDS model (S15; PRJNA610193), which we processed using the same pipeline as our S10 and S30 data (Appendix A). To correlate these epigenetic changes with transcription, we analyzed our accompanying RNA-seq data from the prefrontal cortex after S10, S15 and S30 days of stress [15,23], (PRJNA610193, PRJNA323485), designating genes with a *p*-value < 0.05 as significantly differentially expressed (Appendix A).

### 3.11. Statistical Analysis

The behavioral data exhibited a non-normal distribution, necessitating the selection of nonparametric tests. The behavioral data in one of these analyses were assessed using the Kruskal–Wallis test, with the type of stress as a variable. Pairwise comparisons were made by the Mann–Whitney U test. The statistical analyses were conducted using STATISTICA 8 software. A *p*-value below 0.05 was considered indicative of significance.

## 4. Conclusions

Analyzing SDS of different durations allowed us to identify gene clusters whose regulatory regions show altered nucleosome-scale H3K4me3 peak density. The functional significance of alterations in H3K4me3 density remains unclear, as these changes do not correlate with the duration of stress. Instead, a combination of alterations in various nucleosomal peaks likely plays a more significant role in creating a distinctive H3K4me3 signature in the promoter region. Regrettably, methodological limitations related to the use of bulk prefrontal cortex preclude the evaluation of cell-specific alterations in the H3K4me3 landscape. The development of specific bioinformatics algorithms based on experimental data may in the future make it possible to analyze bulk tissue data profiled with a single-cell resolution. Furthermore, understanding H3K4me3 function also requires integrating other histone marks and DNA methylation profiles. Consequently, undertaking extensive studies in the future will enhance our comprehension of the interplay between epigenetic and transcriptional alterations, thereby advancing our insight into the molecular mechanisms underlying stress-related mental disorders in animal models and their correlation with the mechanisms of depression in humans.

## Figures and Tables

**Figure 1 ijms-27-00018-f001:**
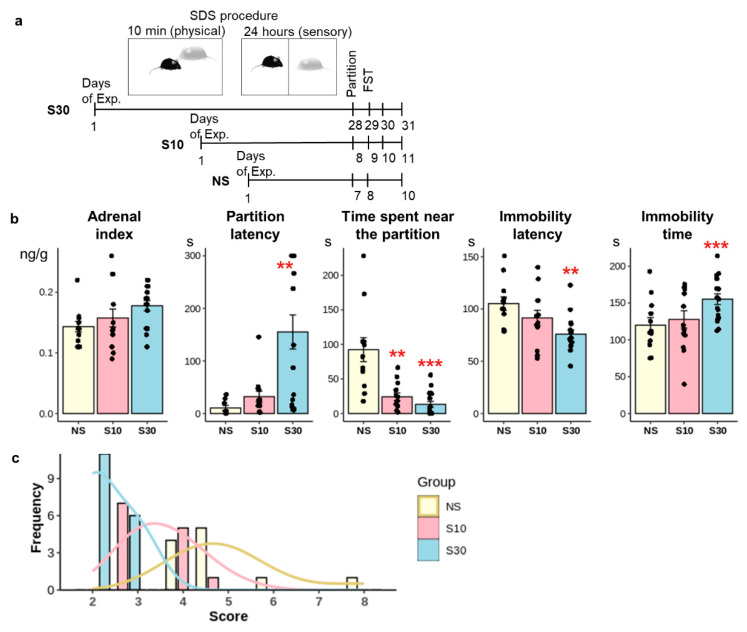
Behavioral disruptions induced by social defeat stress. (**a**). Experimental design. (**b**). Modifications in behavior and physiology; NS group—n = 11; S10 group–n = 13; S30 group n = 17. Data are presented as mean ± standard error of the mean (SEM). **—*p* < 0.01, ***—*p* < 0.001 in comparison to the NS group (control). (**c**). Behavioral assessment for all animals across all groups. Values ranging from 2 to 3 indicate “susceptible” animals, while values exceeding 4 denote “resilient” animals.

**Figure 2 ijms-27-00018-f002:**
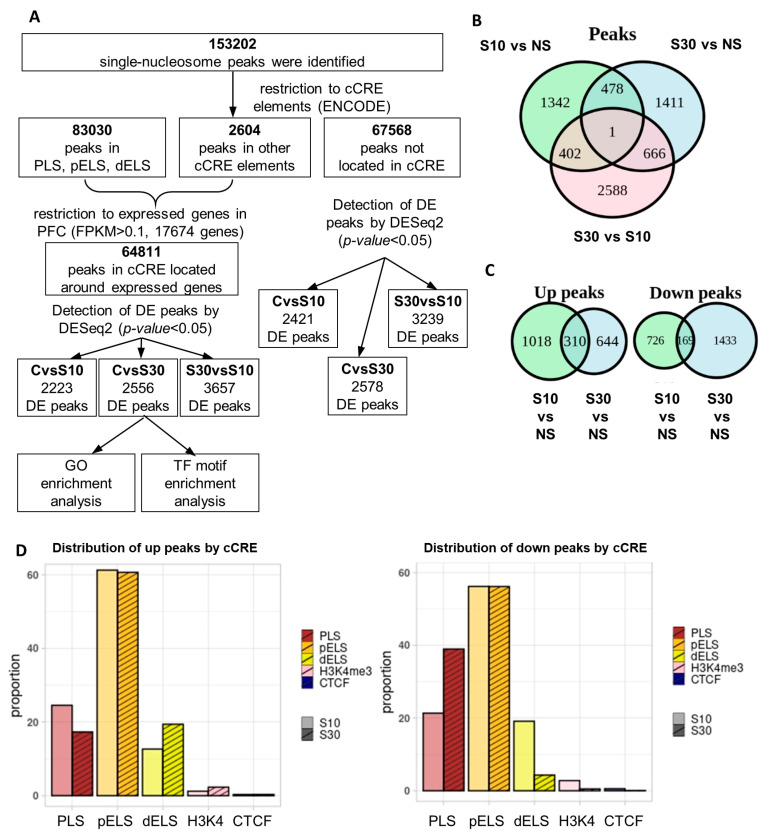
Analysis of H3K4me3 ChIP-seq data (**A**). Flow diagram depicting ChIP-seq data analysis; (**B**). Venn diagram illustrating the comparisons of upregulated and downregulated differential enrichment of H3K4me3 peaks identified in the S10 and S30 groups. (**C**). Venn diagram illustrating H3K4me3 differential enrichment peaks (*p* < 0.05) across various groups; (**D**). Distribution of H3K4me3 peaks based on their localisation and overlap with cCRE (ENCODE).

**Figure 3 ijms-27-00018-f003:**
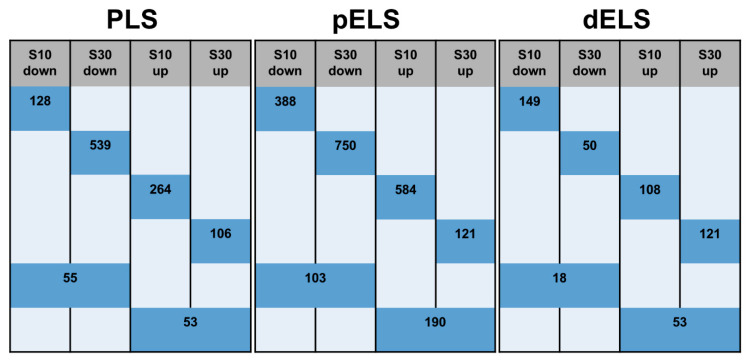
The number of cCREs with up and down H3K4Me3 peaks depending on the duration of stress. The figure shows the numbers of H3K4me3 DE peaks in different groups, which are located in regulatory regions (PLS, pELS, dELS). Merged cells show H3K4me3 DE peaks that change in both groups.

**Figure 4 ijms-27-00018-f004:**
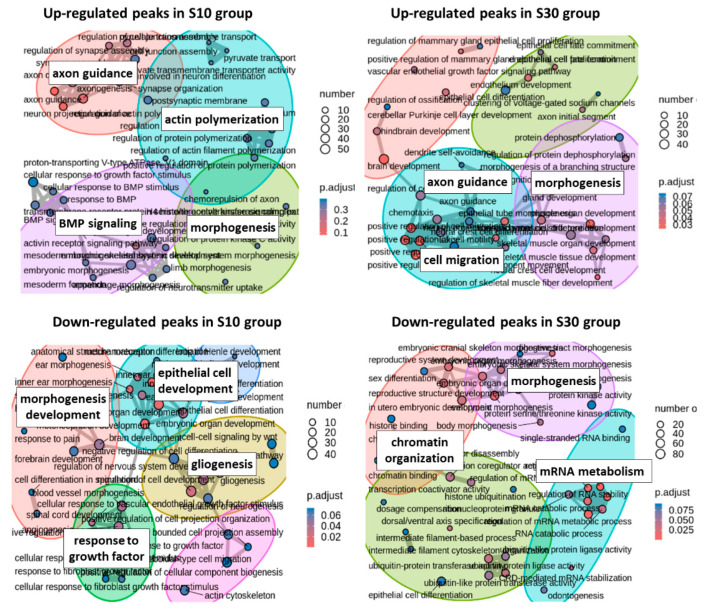
GO terms for genes with up- and down-regulated peaks in the S10 and S30 groups.

**Figure 5 ijms-27-00018-f005:**
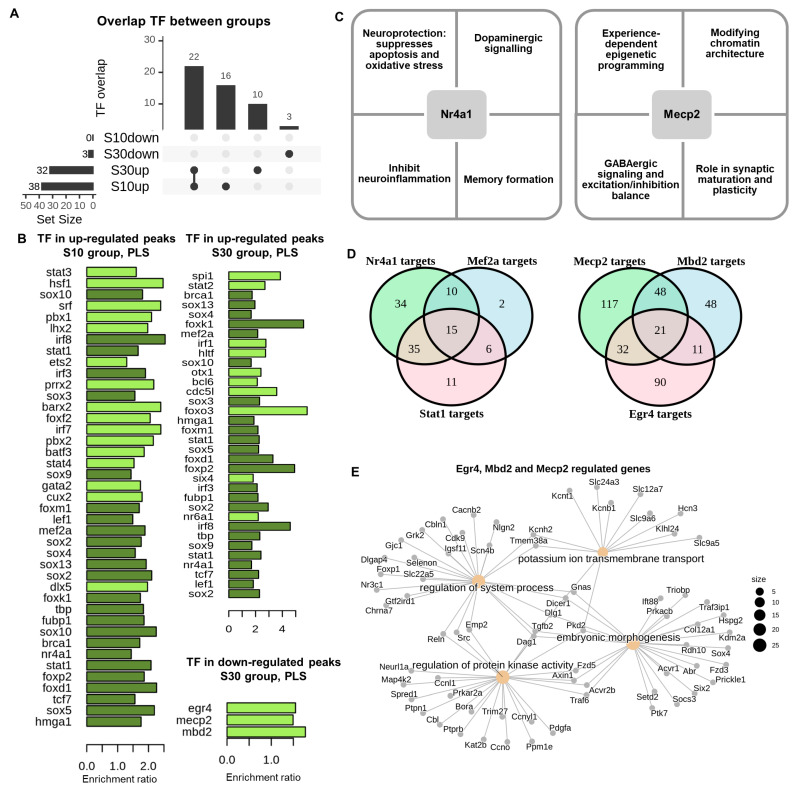
Enrichment analysis of transcription factor binding sites in differentially expressed peaks. (**A**). Overlap between transcription factor sites in S10 and S30 groups. (**B**). Transcription factor sites in upregulated peaks within the S10 and S30 groups. Dark green indicates factors that are enriched in both the S10 and S30 groups. Light green indicates only those factors that are enriched in one of the experimental groups. (**C**). Principal roles of Nr4a1 and Mecp2 transcription factors (**D**). The intersection of Nr4a1, Mef2a, and Stat1 targets and Mecp2, Mbd2, and Egr4 targets within DE peaks (PLS cCRE). (**E**). Clustering of target genes in GO terms of transcription factors Mecp2, Mbd2, and Egr4 within the differentially expressed peaks of the S30 group.

**Figure 6 ijms-27-00018-f006:**
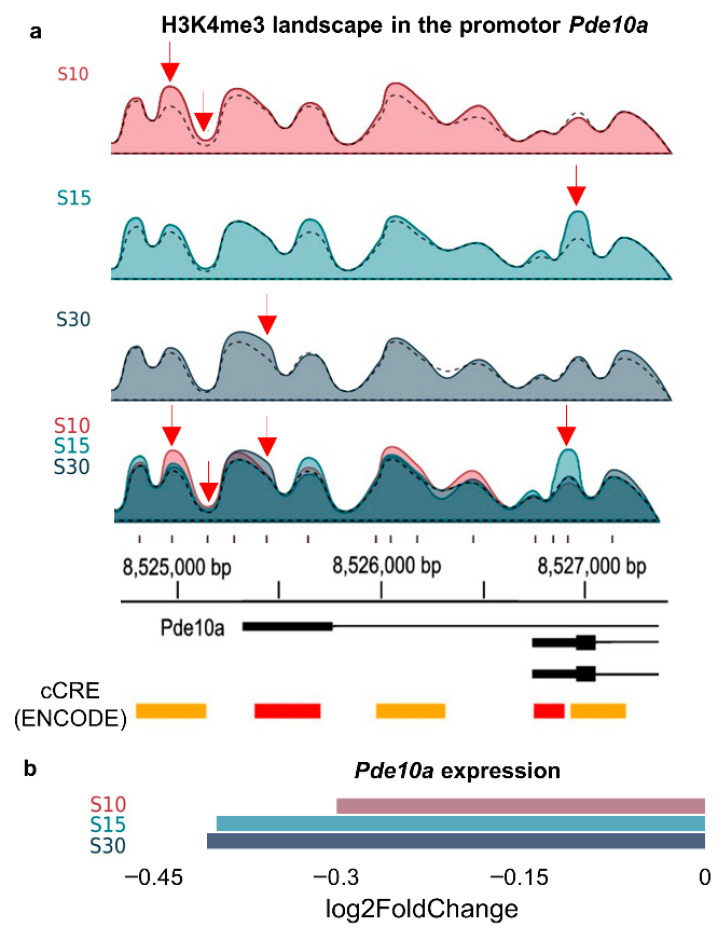
Duration-dependent modifications of the H3K4me3 signature in the promoter of the *Pde10a* gene and its correlation with gene expression. (**a**). Graphical representation of the H3K4me3 landscape in the promoter region of *Pde10a* (derived from IGV browser data, reflecting average values across all samples in a group). The dotted line represents the H3K4me3 landscape of the control group. The red arrow denotes differentially enriched H3K4me3 peaks (*p* < 0.05) in comparison to the control. cCRE—cis-regulatory elements identified by ENCODE; PLS is depicted in red, pELS in yellow. (**b**). The modification of *Pde10a* expression is contingent upon the duration of stress, with *p* < 0.05 for all groups relative to the control.

**Table 1 ijms-27-00018-t001:** The list of single-nucleosome cCRE DE peaks of S30 group compared to S10 group (*p*.adj < 0.1).

Peak Location	log2 FoldChange	*p*.adj	Gene Name	cCRE Type
chr5: 129,584,219–129,584,365	−0.58052	0.047	*Mmp17*	PLS
chr4: 43,875,569–43,875,715	−0.51618	0.047	*Reck*	PLS
chr15: 3,979,351–3,979,497	−0.57101	0.069	*Fbxo4*	PLS
chr15: 93,595,540–93,595,686	−0.61697	0.069	*Prickle1*	pELS
chr9: 75,559,444–75,559,590	−0.51674	0.069	*Tmod3*	PLS
chr5: 147,430,238–147,430,384	−0.49941	0.069	*Pan3*	PLS
chr17: 49,959,957–4,996,103	−0.65811	0.091	*Arid1b*	pELS
chr5: 76,184,229–76,184,375	−0.66802	0.094	*Tmem165*	pELS

**Table 2 ijms-27-00018-t002:** Comparison of genes with DE peaks in its promoter region and DE genes after SDS of various duration.

	Number of DE Genes		
Number of Genes with DE Peaks	S10 (2609 *)	S15 (1201 **)	S30 (2572 *)
S10 (1866)	237 (9.08%)	189 (15.74%)	309 (12.01%)
S15 (1149)	235 (9.07%)	153 (12.74%)	242 (9.40%)
S30 (2127)	258 (9.89%)	186 (15.49%)	351 (13.64%)

* Based on data from [23], *p* value < 0.05 as thresholds. ** Based on data from [15], *p* value < 0.05 as thresholds.

## Data Availability

The original data that support the findings of this study are available from the corresponding author upon reasonable request. Raw data on ChIP-seq were deposited in the NCBI BioProject database under the project ID PRJNA1359597.

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
