# Peer review of "Epigenetic Signatures of Social Defeat Stress Varying Duration"

_ijms, 2025, doi:10.3390/ijms27010018_

Round 1
Reviewer 1 Report
Comments and Suggestions for Authors
The paper entitled "Epigenetic signatures of social defeat stress varying duration" represents a high-quality study in the area of molecular neuroscience. Stress is a primary risk factor for prevalent psychopathologies including affective disorders (depressive and anxiety disorders, PTSD, etc.). It also increases the risk of metabolic, cardiovascular, and neurodegenerative diseases. The extensive negative consequences of chronic stress are associated with molecular and neurochemical alterations in the brain. Nevertheless, despite certain progress in the investigation of stress-induced psychopathologies over the last twenty years, there are still gaps in our knowledge of molecular mechanisms underlying their development.
Chronic social defeat stress (SDS) is well validated and widely used paradigm to produce stress-induced depressive-like state in mice. The research group of Dr. Bondar is well known in the field for investigating the molecular mechanisms of SDS. This study was aimed at examining the epigenetic mechanisms of stress response, namely, the role of histone methylation. Here, they studied the impact of SDS of different durations (10 and 30 days) on the genome-wide H3K4me3 (trimethylation at the 4th lysine residue of histone H3) landscape in the prefrontal cortex of C57BL/6. The study involved both up-to-date molecular methods (ChIP-seq) and bioinformatical analysis that are verified and widely used in the field and in previous studies of the group published in highly impacted peer-reviewed journals. Although it is not a breakthrough, the study by Bondar et al. is a good quality research that broadens our knowledge about epigenetic mechanisms of stress response regulation in the prefrontal cortex. The paper is well written and illustrated. The conclusions are accurate and supported by the results. Noteworthy, the authors clearly realize limitations of their approach and correctly indicated them in the paper. I truly endorse this paper for publication in IJMS and consider it as interesting for readership of the journal.
Minor point that I would recommend to authors to improve the manuscript:
please, decipher the abbreviation (and add to Abbreviations) ‘H3K4me3’ and add some information about this modification into Introduction (about its significance and role in gene regulation in general and its relation to stress and stress-induced disturbances in particular, etc.). Otherwise, its is hard to perceive and estimate your results as separate independent study as a reader needs to refer to your previous research to get the main idea of your work.
Author Response
Dear Editor and Referees:
Thank you for allowing us to submit a revised draft of our manuscript titled “Epigenetic signatures of social defeat stress varying duration”. We are grateful for your review of the manuscript and your valuable comments and concerns. We have been able to incorporate into the manuscript most of the suggestions provided by the reviewers.
We have highlighted the revisions within the manuscript.
Below, marked in red, are point-by-point responses to the reviewers’ comments and concerns.
Sincerely,
Natalia Bondar and Vasiliy Reshetnikov
The paper entitled "Epigenetic signatures of social defeat stress varying duration" represents a high-quality study in the area of molecular neuroscience. Stress is a primary risk factor for prevalent psychopathologies including affective disorders (depressive and anxiety disorders, PTSD, etc.). It also increases the risk of metabolic, cardiovascular, and neurodegenerative diseases. The extensive negative consequences of chronic stress are associated with molecular and neurochemical alterations in the brain. Nevertheless, despite certain progress in the investigation of stress-induced psychopathologies over the last twenty years, there are still gaps in our knowledge of molecular mechanisms underlying their development.
Chronic social defeat stress (SDS) is well validated and widely used paradigm to produce stress-induced depressive-like state in mice. The research group of Dr. Bondar is well known in the field for investigating the molecular mechanisms of SDS. This study was aimed at examining the epigenetic mechanisms of stress response, namely, the role of histone methylation. Here, they studied the impact of SDS of different durations (10 and 30 days) on the genome-wide H3K4me3 (trimethylation at the 4th lysine residue of histone H3) landscape in the prefrontal cortex of C57BL/6. The study involved both up-to-date molecular methods (ChIP-seq) and bioinformatical analysis that are verified and widely used in the field and in previous studies of the group published in highly impacted peer-reviewed journals. Although it is not a breakthrough, the study by Bondar et al. is a good quality research that broadens our knowledge about epigenetic mechanisms of stress response regulation in the prefrontal cortex. The paper is well written and illustrated. The conclusions are accurate and supported by the results. Noteworthy, the authors clearly realize limitations of their approach and correctly indicated them in the paper. I truly endorse this paper for publication in IJMS and consider it as interesting for readership of the journal.
Minor point that I would recommend to authors to improve the manuscript:
please, decipher the abbreviation (and add to Abbreviations) ‘H3K4me3’ and add some information about this modification into Introduction (about its significance and role in gene regulation in general and its relation to stress and stress-induced disturbances in particular, etc.). Otherwise, its is hard to perceive and estimate your results as separate independent study as a reader needs to refer to your previous research to get the main idea of your work.
Reply: Thank you for your appreciation of the manuscript. The text has been adjusted according to your recommendations.
Reviewer 2 Report
Comments and Suggestions for Authors
This study highlights the changes in the chromatin organization with a higher level of stress that is closely related to depression like phenotypes.
I have a few comments below:
- The authors should show values for each sample rather than only summary statistics in the graphs for Figure 1.
- The manuscript should clearly state:
- The total number of mice initially used in the study, and how many mice were subjected to the 30-day and 10-day stress paradigms.
- Whether any mice died during the paradigm.
- How many mice were used for behavioral assays?
- How many mice exhibited depression-like behavior vs. how many mice were resilient?
- Whether the same cohort was also used for molecular analyses or if separate groups were used. How did the authors identify which mouse to use for the ChIP experiments?
- Figures need improved clarity and labeling.
- Figure legends should be more descriptive to allow the figures to be understood independently of the main text.
- In Fig. 1, multiple graphs appear to be merged into a single panel. Each plot should be presented as a separate panel with clear labels.
- In Fig. 2B, the Venn diagram should be properly labeled with the sample groups it represents.
Author Response
Dear Editor and Referees:
Thank you for allowing us to submit a revised draft of our manuscript titled “Epigenetic signatures of social defeat stress varying duration”. We are grateful for your review of the manuscript and your valuable comments and concerns. We have been able to incorporate into the manuscript most of the suggestions provided by the reviewers.
We have highlighted the revisions within the manuscript.
Below, marked in red, are point-by-point responses to the reviewers’ comments and concerns.
Sincerely,
Natalia Bondar and Vasiliy Reshetnikov
Reviewer 2
This study highlights the changes in the chromatin organization with a higher level of stress that is closely related to depression like phenotypes.
I have a few comments below:
- The authors should show values for each sample rather than only summary statistics in the graphs for Figure 1.
- The manuscript should clearly state:
- The total number of mice initially used in the study, and how many mice were subjected to the 30-day and 10-day stress paradigms.
- How many mice were used for behavioral assays?
- How many mice exhibited depression-like behavior vs. how many mice were resilient?
- Whether the same cohort was also used for molecular analyses or if separate groups were used. How did the authors identify which mouse to use for the ChIP experiments?
Reply: Thank you for highlighting your emphasis. Figure 1 has undergone a complete redesign. Individual values have been assigned to each animal in the behavioral assessments. Furthermore, we have incorporated the clustering of animals according to behavioral scores derived from test results (partition test and Porsolt’ test). Our results align with the study conducted by Krishnan et al. 2007. Furthermore, we have included data on the quantity of animals within groups in the caption of Figure 1. For molecular investigations, we utilized animals exhibiting the most distinctive behaviors for each category.
- Whether any mice died during the paradigm.
Reply: No, the mice did not die during the experiment. The model focuses specifically on psycho-emotional stress and does not allow for serious physical harm to the animals. In cases of overly aggressive confrontation, the researcher terminated the experiment early (less than 10 minutes). Furthermore, overly aggressive aggressors were excluded from the experiment.
- Figures need improved clarity and labeling.
- Figure legends should be more descriptive to allow the figures to be understood independently of the main text.
Reply: Thank you. The figure legends have been revised.
- In Fig. 1, multiple graphs appear to be merged into a single panel. Each plot should be presented as a separate panel with clear labels.
Reply: Thanks for your remark. Figure 1 has undergone a complete redesign.
- In Fig. 2B, the Venn diagram should be properly labeled with the sample groups it represents.
Reply: We adjusted the Figure 2.
Reviewer 3 Report
Comments and Suggestions for Authors
In this article, Bondar et al. evaluated the epigenetic effect of the most ethologically valid animal model of depression, chronic social defeat stress (SDS), on the prefrontal cortex in mice. They focused on the H3K4me3 genomic landscape using ChIP-Seq and observed that the density of H3K4me3 peaks in candidate cis-regulatory elements (cCREs) of genes expressed in the prefrontal cortex increased with the duration of SDS. Nevertheless, the difference associated with increased stress duration is not completely unidirectional and may reflect bidirectional dysregulation of the transcriptional mechanism. Furthermore, comparison of H3K4me3 peaks with RNA-Seq data revealed a lack of correlation between the epigenetic mark and gene expression level. Overall, the manuscript is well written, and the topic is of great interest due to the lack of precise knowledge about the molecular mechanisms underlying stress-induced affective disorders. Nevertheless, we would like the authors to address certain points in order to fully appreciate the novelty of their work.
Major concerns:
- In the first paragraph of the “Results” section, the authors did not explicitly specify the number of animals tested for each group. Since we cannot assess the individual behavior of each animal, we do not know whether some animals are resistant to the SDS procedure. We believe this is a major limitation of the study as a whole and feel that a significantly improved analysis should consider exploring the epigenetic signature of susceptibility versus resilience. Indeed, the fact that mice in the S10 group do not differ from controls in terms of adrenal index, partition latency, immobility latency, and immobility time suggests a high rate of resilience. The authors should definitely discuss the proportion of resilience in their hands compared to previous published studies using the same protocol, particularly in relation to the seminal article by Krishnan V et al published in Cell in 2007. In line 99, the authors cite the work of Lu J et al, who evaluated the effect of prolonged stress on resilience and consistency in depressive-like behaviors. Therefore, when the authors state that “the data are consistent with other studies...”, we really need to see to what extent the proportion of resilience is also consistent with previous reports from different researchers. Lines 106-109: The authors acknowledge heterogeneity observed after 10 days, but it is clear that the data presented do not allow this heterogeneity to be observed.
- Lines 134-138: What are the eight peaks that varied in the S30 group compared to the S10 group after correction for multiple comparisons, and why not highlight these peaks?
- Lines 167-169: the authors suggest that the absence of significant epigenetic changes is related to the use of bulk prefrontal cortical tissue, but what about “contamination” by resilience?
- For Figure 5 A: the bar graphs are difficult to read because the transcriptional factors are not ranked according to enrichment rate. Why do we also see two different shades of green? The authors indicate that the key regulator identified is Nr4a1, but it appears that the highest peak in S10 is for Hsf1: a very interesting stress factor also involved in chromatin remodeling. We believe the authors should also elaborate on this point. And what about the highest peak in S30 for Foxo3?
- The authors have failed to highlight the limitations of their work regarding the sample size for ChIP-Seq experiments and the lack of validation on specific genes.
Minor concerns:
-In the abstract: line 28, correct the typo, a space is missing: ... a better understanding of the dynamics...
-In the abstract, last sentence, a verb is missing: did the authors mean to say “these results provide a better understanding of...”?
- In the introduction, lines 64-66, the authors should specify that the initial study was conducted on mice raised normally or subjected to early maternal separation: otherwise, it appears that the current study uses data obtained in a previous study without any real new experiments.
- The authors did not indicate in the Methods section how they assessed the adrenal index; please add a specific procedure and a reference.
- For Figure 1, it would be important to include the number of animals tested in the caption. Alternatively, instead of a bar graph, a scatter plot would allow for a better appreciation of individual variability and help to understand why, for partition latency, there is no significant difference between the NS group and the S30 group.
- In Supplementary Table 1, it would be useful to rank the peaks according to the p-value so that the best results can be seen immediately.
- For Figure 2D, the color used to identify the group is not clear: we suggest using a solid color for S10 and a dotted color for S30.
-For Figure 3: what do the second set of numbers at the bottom of the panels represent?
-Lines 241-242: the authors mention that axonal guidance is common to the S10 and S30 groups, but do not mention morphogenesis. Why?
Author Response
Dear Editor and Referees:
Thank you for allowing us to submit a revised draft of our manuscript titled “Epigenetic signatures of social defeat stress varying duration”. We are grateful for your review of the manuscript and your valuable comments and concerns. We have been able to incorporate into the manuscript most of the suggestions provided by the reviewers.
We have highlighted the revisions within the manuscript.
Below, marked in red, are point-by-point responses to the reviewers’ comments and concerns.
Sincerely,
Natalia Bondar and Vasiliy Reshetnikov
Reviewer 3
In this article, Bondar et al. evaluated the epigenetic effect of the most ethologically valid animal model of depression, chronic social defeat stress (SDS), on the prefrontal cortex in mice. They focused on the H3K4me3 genomic landscape using ChIP-Seq and observed that the density of H3K4me3 peaks in candidate cis-regulatory elements (cCREs) of genes expressed in the prefrontal cortex increased with the duration of SDS. Nevertheless, the difference associated with increased stress duration is not completely unidirectional and may reflect bidirectional dysregulation of the transcriptional mechanism. Furthermore, comparison of H3K4me3 peaks with RNA-Seq data revealed a lack of correlation between the epigenetic mark and gene expression level. Overall, the manuscript is well written, and the topic is of great interest due to the lack of precise knowledge about the molecular mechanisms underlying stress-induced affective disorders. Nevertheless, we would like the authors to address certain points in order to fully appreciate the novelty of their work.
Major concerns:
- In the first paragraph of the “Results” section, the authors did not explicitly specify the number of animals tested for each group. Since we cannot assess the individual behavior of each animal, we do not know whether some animals are resistant to the SDS procedure. We believe this is a major limitation of the study as a whole and feel that a significantly improved analysis should consider exploring the epigenetic signature of susceptibility versus resilience. Indeed, the fact that mice in the S10 group do not differ from controls in terms of adrenal index, partition latency, immobility latency, and immobility time suggests a high rate of resilience. The authors should definitely discuss the proportion of resilience in their hands compared to previous published studies using the same protocol, particularly in relation to the seminal article by Krishnan V et al published in Cell in 2007. In line 99, the authors cite the work of Lu J et al, who evaluated the effect of prolonged stress on resilience and consistency in depressive-like behaviors. Therefore, when the authors state that “the data are consistent with other studies...”, we really need to see to what extent the proportion of resilience is also consistent with previous reports from different researchers. Lines 106-109: The authors acknowledge heterogeneity observed after 10 days, but it is clear that the data presented do not allow this heterogeneity to be observed.
Reply: Thank you for highlighting your emphasis. Figure 1 has undergone a complete redesign. Individual values have been assigned to each animal in the behavioral assessments. Furthermore, we have incorporated the clustering of animals according to behavioral scores derived from test results (partition test and Porsolt’ test). Our results align with the study conducted by Krishnan et al. 2007. Furthermore, we have included data on the quantity of animals within groups in the caption of Figure 1. For molecular investigations, we utilized animals exhibiting the most distinctive behaviors for each category.
- Lines 134-138: What are the eight peaks that varied in the S30 group compared to the S10 group after correction for multiple comparisons, and why not highlight these peaks?
Reply: Thanks for your remark. We have expanded the description of these results and added Table 1 with a description of these peaks.
- Lines 167-169: the authors suggest that the absence of significant epigenetic changes is related to the use of bulk prefrontal cortical tissue, but what about “contamination” by resilience?
Reply: The selection of animals for molecular studies was conducted through evaluation of behavioral outcomes. Consequently, we suggest that the variability of behavioral repertoires does not significantly influence the results achieved.
- For Figure 5 A: the bar graphs are difficult to read because the transcriptional factors are not ranked according to enrichment rate. Why do we also see two different shades of green?
Reply: Dark green indicates factors that are enriched in both the S10 and S30 groups. Light green indicates only those factors that are enriched in one of the experimental groups. We adjusted a figure legend.
The authors indicate that the key regulator identified is Nr4a1, but it appears that the highest peak in S10 is for Hsf1: a very interesting stress factor also involved in chromatin remodeling. We believe the authors should also elaborate on this point. And what about the highest peak in S30 for Foxo3?
Reply: Thank you for your comment. Initially, we resolved to delineate the transcription factors exhibiting enrichments in both the S10 and S30 cohorts. In response to your remark, we have expanded upon the possible roles of the transcription factors Hsf1 and Foxo3.
- The authors have failed to highlight the limitations of their work regarding the sample size for ChIP-Seq experiments and the lack of validation on specific genes.
Reply: We entirely agree with these comments. Limitations on the interpretation of study findings have been incorporated into Section 2: Results and Discussion.
Minor concerns:
-In the abstract: line 28, correct the typo, a space is missing: ... a better understanding of the dynamics...
-In the abstract, last sentence, a verb is missing: did the authors mean to say “these results provide a better understanding of...”?
Reply: Thank you. The text has been amended.
- In the introduction, lines 64-66, the authors should specify that the initial study was conducted on mice raised normally or subjected to early maternal separation: otherwise, it appears that the current study uses data obtained in a previous study without any real new experiments.
Reply: Thank you for your comment. Indeed, in the Introduction, we referenced our previous work on the effect of 15 days of social defeat stress on H3K4me3 distribution in mice reared normally or subjected to maternal separation early in life. We have added this clarification to the text. In this paper, we compare our new data for 10 and 30 days of stress with previous H3K4me3 data obtained in a group of normally reared mice with 15 days of social defeat stress.
“Our group previously performed the initial study assessing the effects of a 15-day chronic SDS on the genome-wide H3K4me3 landscape in the prefrontal cortex of C57BL/6 mice subjected to maternal separation in early in life [15]”
- The authors did not indicate in the Methods section how they assessed the adrenal index; please add a specific procedure and a reference.
Reply: Thank you for your comment. Mice and adrenal glands were weighted and adrenal indices were calculated (adrenal mass (ng)/body mass (g)). We added this description to the Methods
- For Figure 1, it would be important to include the number of animals tested in the caption. Alternatively, instead of a bar graph, a scatter plot would allow for a better appreciation of individual variability and help to understand why, for partition latency, there is no significant difference between the NS group and the S30 group.
Reply: The figure has been revised.
- In Supplementary Table 1, it would be useful to rank the peaks according to the p-value so that the best results can be seen immediately.
Reply: Supplementary Table 1 has been updated.
- For Figure 2D, the color used to identify the group is not clear: we suggest using a solid color for S10 and a dotted color for S30.
Reply: Figure 2 has been updated.
-For Figure 3: what do the second set of numbers at the bottom of the panels represent?
Reply: Figure 3 has been updated, and the legend has been amended.
-Lines 241-242: the authors mention that axonal guidance is common to the S10 and S30 groups, but do not mention morphogenesis. Why?
Reply: We apologise for our oversight; the text has been amended.
Round 2
Reviewer 3 Report
Comments and Suggestions for Authors
Thank you for taking all our comments into account. However, there is one minor point remaining:
In your response indicating that the location of the eight significant peaks should be included in the revised manuscript, you mention the Tmem165 gene located on chromosome 5 in the text, but the Ticam1 gene located on chromosome 17 in Table 1: please harmonize this information.
Author Response
Dear reviewer,
I appreciate you reviewing and identifying the typo. The table has been updated.